# RNA Structure in the 5′ Untranslated Region of Enterovirus D68 Strains with Differing Neurovirulence Phenotypes

**DOI:** 10.3390/v15020295

**Published:** 2023-01-20

**Authors:** Connor Eastman, William E. Tapprich

**Affiliations:** Department of Biology, University of Nebraska at Omaha, Omaha, NE 68182, USA

**Keywords:** 5′ untranslated region, enterovirus D68, RNA structure, internal ribosome entry site, neurovirulence

## Abstract

Enterovirus-D68 (EV-D68) is a positive-sense single-stranded RNA virus within the family *Picornaviridae*. EV-D68 was initially considered a respiratory virus that primarily affected children. However, in 2014, EV-D68 outbreaks occurred causing the expected increase in respiratory illness cases, but also an increase in acute flaccid myelitis cases (AFM). Sequencing of 2014 outbreak isolates revealed variations in the 5′ UTR of the genome compared to the historical Fermon strain. The structure of the 5′ UTR RNA contributes to enterovirus virulence, including neurovirulence in poliovirus, and could contribute to neurovirulence in contemporary EV-D68 strains. In this study, the secondary and tertiary structures of 5′ UTR RNA from the Fermon strain and 2014 isolate KT347251.1 are analyzed and compared. Secondary structures were determined using SHAPE-MaP and TurboFold II and tertiary structures were predicted using 3dRNAv2.0. Comparison of RNA structures between the EV-D68 strains shows significant remodeling at the secondary and tertiary levels. Notable secondary structure changes occurred in domains II, IV and V. Shifts in the secondary structure changed the tertiary structure of the individual domains and the orientation of the domains. Our comparative structural models for EV-D68 5′ UTR RNA highlight regions of the molecule that could be targeted for treatment of neurotropic enteroviruses.

## 1. Introduction

Enterovirus D68 (EV-D68) is a positive-sense single-stranded RNA virus of the genus *Enterovirus* and family *Picornaviridae*. When EV-D68 was first discovered in California in 1962 as a childhood infection, it was thought to only cause respiratory illness [1]. However, in 2014 there was an outbreak of EV-D68 infections in the United States, Canada and several European countries, totaling over 2000 infections worldwide [2]. In addition to the rise in respiratory illness cases that would be expected from an outbreak of EV-D68, there was an increase in the incidence of acute flaccid myelitis (AFM) [3,4,5,6]. It was found that during the outbreaks there was a high correlation between the development of AFM and EV-D68 infection and that the neuropathogenic infections tended to occur in individuals infected with the B1 subclade of EV-D68 [3,6,7]. Further, retrospective research and work in animal models have supported a causal relationship between EV-D68 infection and AFM by fulfilling the majority of Bradford Hill criteria and fulfilling Koch’s postulates [8,9,10]. Since 2014, outbreaks of EV-D68 have occurred on a biennial basis, excluding 2020, which may be due to public health responses to the SARS-CoV-2 pandemic, as EV-D68 began to re-emerge as lockdown procedures were loosened in Europe [4,11,12,13].

The question is, what has caused contemporary strains of EV-D68 to cause AFM, which was previously unseen in historical cases? Possible virulence and tropism phenotypic changes that could result in contemporary EV-D68 strains causing increased rates of AFM include the gain of neurotropism, increased neurovirulence and increased neuroinvasion. These changes are not mutually exclusive and could coexist. Work in neuroblastoma cell lines has suggested that contemporary strains of EV-D68 have gained the ability to infect neuronal cells [14], and work in mice as well as neuronal cell cultures has shown increased neuroinvasion in contemporary strains [15]. However, studies in mice, human neuronal cell cultures and organotypic mice tissue cultures have also shown that both historic and contemporary isolates of EV-D68 are capable of infecting neuronal cells, suggesting that both historical and contemporary strains of EV-D68 may be neurotropic [16].

Some general determinants of EV-D68 neurovirulence in later strains have been identified. For example, studies have suggested that viral capsid proteins VP1, VP2 and VP3 and nonstructural proteins 2A, 2C, 3A and 3D have an influence on neurovirulence [6,17]. In addition, there is support for the role of the internal ribosomal entry site (IRES) within the 5′ UTR of the EV-D68 genome as a determinant of neurovirulence [6,17,18]. The 5′ UTR, with its IRES element, is essential for viral translation and replication. The region has been shown to be a major determinant of virulence in several enteroviruses, including poliovirus, coxsackievirus B3 and enterovirus-A71, supporting its potential role in the increased neurovirulence of modern EV-D68 strains [19,20,21].

The 5′ UTR of enteroviruses is a highly structured RNA region that is responsible for cap-independent translation initiation and replication of the enteroviral genome [22,23,24,25]. The IRES was first discovered in encephalomyocarditis virus (EMCV) and poliovirus [22,26]. EV-D68 contains a type 1 IRES, such as poliovirus and coxsackievirus, which encompasses domains II through VI of the six RNA domains within the 5′ UTR [27]. The IRES interacts with ribosomal subunits, eukaryotic initiation factors and IRES trans-acting factors (ITAFs) to carry out the cap-independent translation of the genome [22,26,28].

Our work compares the structures of the 5′ UTR of EV-D68 from the historic Fermon strain and the 2014 isolate KT347251.1. The secondary structure of the 5′ UTR RNA of each strain was determined using SHAPE-MaP and TurboFold II [29]. The tertiary structures were then predicted using 3dRNAv2.0 [30]. Nucleotide differences between the strains of EV-D68 caused significant remodeling of both the secondary and tertiary structures. Although many functional elements of each domain were preserved between strains, each domain underwent secondary structure changes, including notable changes in domain II and a rearrangement that combined domains IV and V to form a super domain. Shifts in secondary structure changed both the tertiary structure of the individual domains and the orientation of the domains in relation to one another.

## 2. Materials and Methods

### 2.1. Design, Construction and Isolation of Plasmids with 5′ UTR Sequences

For both the Fermon (NC_038308.1) and 2014 (KT347251.1) strains, a p2RZ plasmid containing the 5′ UTR sequence was designed, constructed and isolated [31]. The strain-specific 5′ UTR DNA sequences were obtained from Integrated DNA Technologies (Coralville, IA, USA) and cloned into the p2RZ plasmid within the multicloning site. The p2RZ plasmid was chosen as the vector for in vitro transcription since it contains sequences that produce a 5′ hammerhead ribozyme and 3′ hepatitis delta virus ribozyme once it is transcribed into RNA from a T7 promoter. These ribozymes cleave the transcribed RNA product to yield the desired 5′ and 3′ ends. The assembled plasmid was transformed and maintained in Invitrogen Subcloning Efficiency DH5α *Escherichia coli* cells (Carlsbad, CA, USA). Glycerol stocks of transformed cells were stored at −80 °C.

To isolate the p2RZ plasmid constructs, an isolated colony from an LB plate containing 200 ug/mL of ampicillin (LBamp) was inoculated in 25 mL of liquid LBamp and cultured for 14 to 16 h at 37 °C. Plasmid DNA was isolated from 5 mL of this overnight culture using the Qiagen Spin Miniprep Kit (Hilden, Germany) according to the manufacturer’s instructions. Plasmid concentration was evaluated using a Thermo Scientific NanoDrop 2000c while the DNA was visualized on a 1% wt/v agarose gel containing 2 μg/mL of EtBr following electrophoresis at 70 V for 60 min and imaged on an Axygen imager at a wavelength of 302 nm.

### 2.2. Production of 5′ UTR RNA

The plasmids were linearized using Eco-RV (Promega, R6351, Madison, WI, USA) at 37 °C for 180 min and purified by extraction using phenol chloroform and chloroform and ethanol precipitation. The presence of the plasmid was once again confirmed using an agarose gel and imaging process as described above. The linearized DNA was transcribed into RNA using an Invitrogen MEGAscript T7 High Yield Transcription Kit (Waltham, MA, USA) according to manufacturer’s instructions. RNA was isolated by two rounds of phenol extraction using phenol chloroform and chloroform and precipitated using isopropanol precipitation. The presence of RNA was confirmed using a 1.5% wt/v agarose gel with 2 μg/mL of EtBr electrophoresed at 70 V for 120 min and imaged on the Axygen imager. RNA was further purified using an Invitrogen MEGAclear kit, and concentration and purity were determined using a NanoDrop 2000c.

### 2.3. Modification of RNA

SHAPE-MaP experiments were run according to the protocols of Smola et al. [32]. Three RNA samples were prepared for each SHAPE-MaP experiment: a folded and modified sample (positive), a folded and unmodified sample (negative), and a denatured and modified sample (DC). To generate the positive and negative samples, 10 pmol of RNA was diluted into 12 μL using ddH_2_O, denatured at 95 °C for two minutes and cooled on ice for two minutes. The RNA was then refolded at 37 °C for 20 min after the addition of 6 μL of 3.3× folding buffer (33 mM HEPES pH 8.0, 333 mM NaCl, 33 mM MgCl_2_). A 300 mM stock of 1-methyl-7-nitroisatoic anhydride (1M7) (Sigma-Aldrich, St. Louis, MO, USA) in DMSO (Fisher bioreagents, Fair Lawn, NJ, USA) was further diluted to 100 mM of 1M7 in DMSO for the reactions. Each modification reaction was performed in duplicate from this point on. The positive sample was modified by the addition of 1 μL of 100 mM 1M7 to 9 μL of RNA solution and incubation at 37 °C for 85 s. For the negative sample, the same procedure is used with the use of neat DMSO rather than the 1M7 solution. Finally, to modify the DC sample, 10 pmol of RNA was diluted into 3 μL of ddH_2_O and mixed with 5 μL of 100% formamide (appliedbiosystems, HiDi Formamide) and 1 μL of DC buffer (500 mM HEPES, 40 mM EDTA). This mixture was incubated at 95 °C for 1 min to denature the RNA. Finally, 1 μL of 100 mM of 1M7 was mixed with 9 μL of denatured RNA and incubated at 95 °C for 1 min. Once the positive, negative and DC sample sets were generated in duplicate, the two samples were pooled and purified using the Qiagen RNeasy Mini Kit (Hilden, Germany) according to manufacturer’s instructions.

### 2.4. Reverse Transcription of RNA and Sequencing of cDNA

To detect modified positions, the RNA was reverse transcribed and sequenced. Samples were reverse transcribed in duplicate using the Invitrogen Superscript II Reverse Transcriptase and New England Biolabs (Ipswich, MA, USA) NEBNext Ultra II Non-directional RNA Second Strand Synthesis module. Once cDNA was generated, samples were pooled and purified using the Invitrogen Purelink PCR Micro Kit (Carlsbad, CA, USA) and sequenced at the University of Nebraska Medical Center Sequencing Core using Illumina MiSeq paired-end sequencing using a Nextera DNA library preparation kit.

### 2.5. Secondary Structure Determination

The secondary structure of RNA from each virus was determined by running the FastQ files generated by sequencing through Shapemapper-2.1.5 (https://github.com/Weeks-UNC/shapemapper2, accessed on 23 February 2020). Sequencing runs were required to pass a minimum read depth of 2000 before moving forward with structure determination. Successful runs were advanced through TurboFold II and aligned against homologous sequence sets (Table 1). Secondary structures generated by TurboFold II were organized and visualized in StructureEditor (https://rna.urmc.rochester.edu/GUI/html/StructureEditor.html, accessed on 5 March 2022).

### 2.6. Tertiary Structure Prediction

Tertiary structure predictions were carried out by converting the ct file produced by TurboFold II to a dot bracket file using the University of Rochester CTtoDOT webserver (https://rna.urmc.rochester.edu/RNAstructureWeb/Servers/ct2dot/ct2dot.html, accessed on 8 March 2022). Then, a DCA file was generated using the Xiao Lab DCA Web Server (http://biophy.hust.edu.cn/new/DCA, accessed on 8 March 2022) to guide the folding. The dot bracket file and DCA file were uploaded into the 3dRNAv2.0 webserver (http://biophy.hust.edu.cn/3dRNA, accessed on 8 March 2022) and five potential tertiary structures were generated based on distance geometry and energy minimization constraints. Each of the five structures were evaluated based on the energy distribution between nucleotides using 3dRNAscore. The structure with the lowest score, symbolizing the highest probability of accurate base pairing and tertiary interaction, was then chosen for further analysis. The resulting tertiary structures were then visualized and pseudocolored in PyMOL 2.5 (https://pymol.org/2/, accessed on 8 March 2022).

## 3. Results

Overall, there are minimal sequence changes between the two strains of EV-D68, with the 2014 strain 93.95% identical to the prototypic Fermon strain. An alignment of the 5′ UTR sequences for the Fermon and 2014 strains of EV-D68 generated by Clustal Omega [33] is shown in Figure 1. In the region spanning domains I-VI, the 5′ UTR of the 2014 strain carries 35 nucleotide changes when compared to the Fermon reference sequence. Each of the domains contains multiple substitutions. Some domains harbor numerous changes, such as domains II and IV, which contain 7 and 11, respectively. All but one of the substitutions in these domains are transitions that maintain the pyrimidine or purine identity of the nucleotide. The remaining domains have three to four changes; two of these domains have one transversion and two have no transversions. Domain I of the 2014 strain is unique in that one of its base changes is an insertion of a cytosine nucleotide at position 30. There are also changes in the linker regions between the domains, such as four substitutions, including a transversion, within the linker region between domains I and II. Between domain VI and the polyprotein start codon, the substitution frequency in the 2014 strain is much greater, including deletions of 23, 8 and 4 nucleotides. While the overall substitution frequency in the 5′ UTR is low, the changes lead to restructuring of most domains at the secondary level and the reorientation of domains in models of the tertiary structure.

The secondary structures of the 5′ UTR for both the Fermon and 2014 EV-D68 strains were determined using Selective 2′ Hydroxyl Acylation by Primer Extension with Mutational Profiling (SHAPE-MaP) and TurboFold II [29]. Figure 2 shows the complete secondary structures for both strains. Positions that are highly modified on the backbone by 1M7, indicating increased structural flexibility (unpaired), are shaded red. The modification data provide constraints that are combined with sequence comparison in TurboFold II to generate the structures. Three biological replicate structures for each strain of EV-D68 were generated and were, on average, 97.11% and 94.83% identical in base-paired positions within domains I to VI for the Fermon and 2014 strain structures, respectively. Differences in structure past domain VI were excluded from the analysis. Most changes in nucleotide base pairing between replicates were observed within domain I; outside domain I, changes in base pairing were primarily localized to nucleotides leading into or out of single-stranded regions. Tertiary structure models seen in Figure 3 were generated using 3dRNAv2.0, which was chosen as the tertiary structure modeling software due to its ability to fold RNA sequences greater than 500 bp with high accuracy, allowing the entire 5′ UTR to be folded simultaneously [30]. Interactive tertiary structure models are available using the pdb files in the Appendix A. Quality of the folded RNAs were evaluated using 3dRNAscore which has been shown to perform similarly to other RNA structure evaluation programs [34]. The lowest scoring tertiary structure models were chosen for each strain, scoring 26.0792 and 26.1333 for the Fermon strain and 2014 strains, respectively, with no statistically significant differences occurring in the scoring between each generated structure.

### 3.1. Domain I

Domain I is characterized by a cloverleaf structure that is highly conserved in enterovirus 5′ UTRs. The overall cloverleaf structure is conserved in both the Fermon and 2014 strains but a single nucleotide insertion at position 30, as well as two point mutations at positions 62 and 63, have led to structural changes in the secondary and tertiary structures (Figure 4 and Figure 5). The most significant domain I secondary structure change is the truncation of stem-loop b and stem-loop d, accompanied by an increase in single-stranded nucleotides in the junction for the 2014 strain. Even with these changes, key elements of the domain I cloverleaf shown to be important for protein binding are retained, including the apical stem of stem-loop d, which contains a pyrimidine-rich mismatched internal loop and a capping tetraloop. Despite the small number of nucleotide changes between the strains in domain I, the impact on the secondary structure is substantial. For example, stem-loop b is significantly shorter in the 2014 strain, leading to the loss of the internal loop found in the Fermon strain. Similarly, the size of the capping loop of stem-loop b is reduced from seven nucleotides to a tetraloop. The C-rich sequence in the capping loop of the Fermon stem-loop b, which is often present in that location in other enteroviral cloverleaf structures, is shifted to the stem in the 2014 strain.

In contrast to the truncation seen in stem-loop 1b, stem-loop 1c expands from a two-nucleotide stem capped by a tetraloop in the Fermon strain to a four-nucleotide stem capped by an eight-nucleotide loop in the 2014 strain. Both strains, however, display an AGCU sequence in the loop. Stem-loop 1d is truncated in the 2014 strain, resulting in the loss of a stem with a bulge loop. However, as mentioned above, the internal loop of the apical stem-loop is maintained in both the Fermon and 2014 strains, and the tetraloop in stem-loop 1d changes from a CUYG tetraloop to a UNCG tetraloop as a result of a C to U substitution at position 63.

The predicted tertiary structure of domain I also shifts between the Fermon and 2014 strains. As can be seen in Figure 5, the orientation of the three helices that compose domain I shift in their orientation. Within the Fermon strain, the stem-loop 1b and 1d helices are orientated away from the rest of the molecule, while the helix that corresponds to stem-loop 1c is directed back towards domains II and III (Figure 5a). Within the 2014 strain, the helices are orientated so that the stem-loops at the end of stems 1b, 1c and 1d are orientated towards domains VI, III and II, respectively (Figure 5b).

### 3.2. Domain II

Domain II undergoes significant structural change between the Fermon and 2014 strains (Figure 4). The relatively long pyrimidine-rich connecting region separating domains I and II, which is a common feature of enteroviral 5′ UTR structures, is present in both strains. However, the connecting region is much longer in the 2014 strain (46 nucleotides) than in the Fermon strain (18 nucleotides). Both strains show a short section of long-range base pairing between nucleotides in the connecting region and nucleotides downstream of domain VI.

The most significant change in the domain II secondary structure is the shift from a three-stem structure ranging from nucleotide 105–179 in the Fermon strain to a singular stem-loop ranging from nucleotide 136–182 and incorporating a large internal loop in the 2014 isolate. This reorganization results in the loss of structural features in the 2014 strain that were present in the Fermon strain. For example, stem 2a in the Fermon strain is not present in the 2014 strain. Instead, the entire span is part of the connecting region between domain I and domain II. The single domain II stem-loop in the 2014 strain is capped by a tetraloop. Certain structural features in stem-loops 2b and 2c of the Fermon strain are preserved in the 2014 strain. The six-nucleotide AAACCA hairpin loop present in stem 2b in the Fermon strain is also single-stranded in the 2014 strain, but it appears in the 5′ side of an internal loop. Similarly, the eight-nucleotide AGCACUUC hairpin loop present in stem 2c is also single-stranded in the 2014 strain, but it appears in the 3′ side of the internal loop.

Just as the secondary structure changes significantly between the Fermon and 2014 strains, so does the model of the tertiary structure (Figure 5). Within the Fermon strain, all three domain II stem structures are orientated outward from the center of the molecule and towards the stem-loops of domain I. Stem-loops 2a and 2b are the closest to stem-loop 1b, while stem-loop 2c is set further away (Figure 5a). In the 2014 strain, the domain II stem, with an internal loop and the tetraloop at its end, are orientated away from domains I and III and appears in the center of the more globular molecule. However, stem-loop 1d is orientated towards the large internal loop of domain II (Figure 5b).

### 3.3. Domain III

Domain III is nearly identical in secondary structure in the Fermon and 2014 strains of EV-D68 (Figure 2a,b). Both strains show a singular stem-loop of 46 nucleotides and both strains contain two internal loops that occur at the same positions. In the Fermon strain, the symmetric internal loop that begins at position 185 contains three nucleotides. In the 2014 strain, this loop is reduced to a two-nucleotide loop due to the U-to-A substitution at position 187, which produces an additional complementarity with the uracil on the 3′ side. The other symmetric internal loop starting at nucleotide 194 is completely conserved based on position and composition. The other structural difference that occurs between the two strains is the collapse of the six-nucleotide hairpin loop in the Fermon strain into a tetraloop, again as a result of a nucleotide substitution that creates an additional complementarity. The degree of conservation in domain III secondary structure also extends to its tertiary structure. In both the Fermon and 2014 strains, domain III maintains its predominantly helical structure and extends away from the rest of the molecule, with only the stem-loop 1b of domain I approaching it in the 2014 strain (Figure 5a,b).

### 3.4. Domain IV

Domain IV undergoes significant changes between the Fermon and 2014 strains (Figure 6 and Figure 7). Within the 2014 strain, domains IV and V partially combine to form a domain IV/V super domain (Figure 6b). The formation of this super domain causes significant structural remodeling of domains IV and V at the secondary level (Figure 6), which also alters the model tertiary structure (Figure 7). However, most of domain IV’s individual structural elements are conserved between the Fermon and 2014 strains. Domain IV in both strains is closed by a helix involving base pairs that form with the 240 region on the 5′ side. The base of domain IV in both strains is a long, extended complex helix that displays apical stem-loops radiating from a junction loop. This generalized secondary structure for domain IV is common in enteroviruses. Four structural elements in domain IV have been shown to be critical for IRES function in enteroviruses: a single-stranded C-rich sequence near position 300, a single-stranded C-rich sequence near position 340, and two GNRA tetraloops that cap the stem-loops radiating from the upper junction loop. All of these elements are present in domain IV of both strains.

Whereas domain IV in the two strains shows common elements that are important for IRES function, several details of the long, extended helix differ markedly between the strains. Most prominently, two stem-loops that are generally grouped in domain V, and do indeed appear in domain V in the Fermon strain, have been subsumed into domain IV in the 2014 strain. This creates a IV/V super domain in the 2014 strain that greatly extends the length of domain IV.

As would be expected, the formation of the IV/V super domain significantly impacts the model of the tertiary structure of the domain IV elements (Figure 7). Domain IV of the Fermon strain extends away from the majority of the molecule and has stems 4a and 4c associated with stem 5d of domain V. However, in the 2014 strain, domain IV takes on a different conformation, with stems 4b and 4c located closer to domains II and III, while stem 4a is located near the IDRR and stem 5c of domain V. Despite this, stems 4b and 4c are still separated from the rest of the molecule.

### 3.5. Domain V

The structure of domain V in the 2014 strain is significantly impacted by the formation of the domain IV/V Super domain. In the Fermon strain, domain IV ends near position 440, which begins the transition into domain V. In the 2014 strain, position 440 is part of the stem-loop we label as 5a. A second stem-loop, labeled as 5b, as well as nucleotides that form the closing helix of the super domain, are part of the expanded domain IV. This leads to an overlapping of domains IV and V, causing a significant expansion of both domains in the 2014 strain.

A three-stem-loop organization is present in domain V of both strains but undergoes significant remodeling. Stem 5a shifts from nucleotide 450 in the Fermon strain to nucleotide 414 in the 2014 strain and shares minimal sequence similarity between the two strains. Stem 5a is mostly composed of the nucleotides that make up the 3′ end of the initial extended helix of domain IV in the Fermon strain. Stem 5b in the 2014 strain is composed of the sequence that makes up the end of stem 5a and beginning of stem 5b of the Fermon strain. Because of this, the internal stem-loop present in stem 5b of the 2014 strain at position 459 maintains the AAU nucleotides present in the hairpin loop at the end of stem 5a in the Fermon strain. Stem 5c is shortened from 24 nucleotides in the Fermon strain to 15 nucleotides in the 2014 strain, and the hairpin loop at the end of the stem is reduced from nine to three nucleotides; however, the UUG motif of the nine-nucleotide loop is preserved in the three-nucleotide loop of the 2014 strain.

As would be expected, the secondary structure changes described above led to significant shifts in the tertiary structure predicted by 3dRNAv2.0 (Figure 7). The movement of the stems 5a and 5b into the domain IV/V super domain also significantly shifts the structure as both stems extend outward and away from the rest of the molecule. The tertiary structure of stem 5c is relatively conserved across the two strains as it is a short helix that extends away from the molecule and does not come close to other domains or stems.

### 3.6. Domain VI

Domain VI is highly conserved between the Fermon and 2014 strains (Figure 2). It occupies the same number of nucleotides and is in the same position with only a slight expansion of the second internal loop at position 586 and an A-to-G substitution at position 600. At the tertiary level, the model shows the Fermon strain domain VI separated from the rest of the molecule and located close to domain IV. However, in the 2014 predicted structure, domain VI is not separated as much from the rest of the molecule and is most closely associated with domain I (Figure 7).

### 3.7. Intrinsically Disordered RNA Region

The intrinsically disordered RNA region (IDRR) of the IRES is the most highly conserved region of the IRES, maintaining a length of 19 and 21 nucleotides in the Fermon and 2014 strains, respectively, and is flanked by domains V and VI. There are no point mutations within this feature of the IRES, making it nearly identical in each strain. It is very similar in the model tertiary structure between the two strains as it is completely disordered and positioned on the outside of the molecule in both. However, it is closely associated with stem 4a of domain IV in the 2014 strain.

## 4. Discussion

Enteroviruses are ubiquitous and prevalent infectious agents with an incredible diversity of viral types, disease manifestations, tissue tropisms and virulence phenotypes [35,36,37]. Despite this diversity, all share a genome organization that includes a long and highly structured 5′ UTR that features six RNA domains: an initial domain called the cloverleaf that forms a three-hairpin junction loop (domain I), followed by five domains that form a type 1 IRES (domains II–VI) [38]. Structural models for this domain organization have been supported by experimental evidence for a few representative enterovirus types, such as poliovirus (PV) [39], coxsackievirus B3 (CVB3) [40,41] and enterovirus A71 (EV-A71) [42]. In some cases, comparative studies of viral strains with varying virulence phenotypes have established RNA structures associated with virulence [43,44]. Even so, relatively few enteroviruses have experimentally supported 5′ UTR RNA structures, and few studies have established direct comparative relationships between enterovirus virulence phenotypes and 5′ UTR RNA structures. In our study, we present an experimentally supported model for the structure of the 5′ UTR from EV-D68 and compare secondary and tertiary structures in strains that differ in neurovirulence. Analysis using SHAPE-MaP, TurboFold II and 3dRNAv2.0 shows that 5′ UTR secondary and tertiary structures of EV-D68 have undergone significant remodeling between the Fermon strain primarily responsible for respiratory disease and the 2014 strain responsible for outbreaks of acute flaccid myelitis.

The secondary structure of every 5′ UTR domain changes in some fashion in the comparative models proposed here. Some domains, such as domains I, III and VI, experience minor changes, while other domains, such as domains II, IV and V, undergo significant structural changes. In the tertiary models predicted by 3dRNAv2.0, the secondary structure differences produce differing overall 5′ UTR conformations. In the 2014 strain, domains I, II, III, V and VI, together with the IDRR, are collected into a compact globular arrangement, with domain IV and the domain IV/V super domain extending away from the globular region, whereas the Fermon strain displays a more extended conformation where domains I, II and III are together on one end and domains IV, V and VI with the IDRR are together on the other. The changes in 5′ UTR structure and organization may contribute to increased neurovirulence in contemporary strains of EV-D68 by improving the efficiency of translation and/or replication in neural tissue.

Our models of the 5′ UTRs of the EV-D68 Fermon and 2014 strains both show common features of enterovirus genome organization. The genome begins with a highly conserved cloverleaf structure in domain I. The cloverleaf is known to play a role in both the translation and replication of the viral genome in enteroviruses [45,46,47,48]. Domain I undergoes shifts in both its secondary and tertiary structure between the Fermon and 2014 isolates. In the Fermon strain, the cloverleaf recapitulates the secondary structure found in PV, CVB3 and EV-A71. In the 2014 strain, domain I maintains its cloverleaf structure, but the junction loop that joins each stem of the cloverleaf is expanded and each of the stem lengths are altered. However, these changes may not be as drastic as they might first appear. It has been shown that the single-stranded regions of domain I, particularly the loop regions, are the functional components of the domain and that the stems can be altered without impacting the function of the domain [47]. Within the capping loops of stems 1b and 1c, as well as the internal loop within stem 1d, the sequences are largely conserved. One exception is the C-rich tract in 1b that binds PCBP2 [46,49]. In the 2014 strain, this region has been absorbed into the stem of stem-loop 1b, which could decrease the interactions between stem-loop 1b and PCBP2 and alter the balance between translation and replication [50]. The internal loop within stem-loop 1d is identical within both strains; however, the capping tetraloop, which recruits viral protein 3CD, transitions from a CUYG tetraloop to a UNCG tetraloop in the 2014 strain, which matches more closely with the capping tetraloop in poliovirus [46,48]. It is possible that this change to a more poliovirus-like sequence allows for better recruitment of 3CD to domain I, again influencing the switch from viral translation to viral genome replication.

Changes in the secondary structure also have the potential to impact the functions of the 5′ UTR by changing the orientation of the domains in relation to one another. In the Fermon strain, stem-loop 1d is orientated away from the rest of the molecule. However, within the 2014 isolate, stem-loop 1d is orientated towards domain II and above the oligo(rC) tract that binds PCBP2 in the connecting region between domains I and II. PCBP2 binding to the connecting region oligo(rC) helps to recruit 3CD to stem-loop d [51]. This shift in the orientation of stem 1d could increase the efficiency of viral genome replication as stem 1d is better able to recruit 3CD [45,46,52].

Both strains maintain the long single-stranded pyrimidine-rich connecting region containing the oligo(rC) tract between domains I and II that is common to enteroviruses. Domain II has been previously suggested as a determinant of virulence in other enteroviruses such as CVB3, poliovirus and EV-A71 [20,21,53,54,55,56]; therefore, significant remodeling of domain II is in keeping with the theory that contemporary isolates of EV-D68 are more neurovirulent than their ancestors [21,55,57,58]. The simplification of domain II from a three-stem structure in the Fermon strain to a single stem with a large internal loop in the 2014 strain potentially reveals the aspects of domain II that are essential for function. The hairpin loops at the ends of stems 2b and 2c are conserved as single-stranded regions in the internal loop of domain II in the 2014 strain. The sequence on the 5′ end of the internal loop and in stem-loop 2b contains the conserved enteroviral CCA motif that interacts with hnRNP A1 to promote viral translation [59,60,61,62,63]. In the model of the 2014 tertiary structure, the internal loop of domain II is orientated towards stem 1d of domain I. Bringing these structures closer together could influence virulence in the contemporary EV-D68 strains.

Domains IV and V, as well as the pyrimidine-rich IDRR in the connecting region downstream of domain V, are arguably the most active regions of the type 1 IRES. Binding sites for canonical initiation factors, noncanonical IRES trans-acting factors (ITAFs) and the small ribosomal subunit are clustered in these regions [28,64,65,66,67,68,69]. Structural features of the regions are largely preserved in both Fermon and 2014 strains. Despite forming a super domain by absorbing portions of domain V, the overall secondary structure of domain IV, consisting of a long complex helix topped by a junction loop, is constant between the two strains, with stems 4b and 4c being nearly identical. The only major difference between the Fermon and 2014 strains within domain IV itself is the relocation of stem-loop 4a. Regions that bind factors are conserved within the strains, such as the apical bulge loop of domain IV that interacts with PCBP2, although there is a slight increase in the size of the loop in the 2014 strain that brings the sequence closer to that of poliovirus. This expansion could indicate evolution towards a more neurovirulent, polio-like phenotype through an increased affinity for factors such as PCBP2 [66]. This pattern of drift towards the poliovirus sequence in functional regions carries through to the GNRA tetraloop that caps stem-loop 4b, which has been shown to mediate RNA/RNA interactions within the IRES [70].

The positioning of domain IV within the tertiary structure models of the Fermon and 2014 strains shifts drastically. Within the Fermon strain, domain IV is largely separate from the rest of the molecule, and only associated with domain VI, meaning it must travel greater distances to interact with other portions of the IRES and the factors resident there. In both the Fermon and 2014 models, stem-loop 4b and the apical bulge loop extend away from the molecule potentially available to recruit incoming initiation factors and ITAFs. In the 2014 model, domain IV is closer to the majority of the domains in the molecule as a whole, particularly domains I through III, thereby reducing the distance between these domains and potentially increasing the efficiency of viral translation. This increased kinetic efficiency of domain IV is also supported by work previously performed by Furuse et al., which shows that exchanging domain IV sequences between the Fermon strain and a contemporary EV-D68 strain increases IRES activity in neuronal glioblastoma cells [18].

Domain V recruits factors such as eIF4G, eIF4A and PTB, which in turn aid in the recruitment of ribosomal subunits to the IDRR. Additionally, domain V has been shown to be a determinant of virulence in poliovirus [28,69,71,72,73,74]. Domain V undergoes significant remodeling between the Fermon and 2014 strains at both the secondary and tertiary structure levels. Domain V can be separated into two sections for our analysis. Section one contains stem-loops 5a and 5b, which are not highly conserved between the two strains of EV-D68 as they are absorbed into the domain IV/V super domain in the 2014 strain. Stem-loop 5c, on the other hand, is more highly conserved and experiences fewer changes in its secondary structure. The binding site for eIF-4G has been mapped to regions of domain V that include nucleotides around positions 470, 520, 530 and 540 in other enteroviruses [71,75]. The binding regions include both base-paired and single-stranded nucleotides. These regions are also a mixture of single-stranded and double-stranded nucleotides in both EV-D68 strains, but the 470 region is part of the IV/V super domain in the 2014 strain.

The tertiary structure model shows that the orientation of stem-loop 5c is conserved in both strains. However, the inclusion of stem-loops 5a and 5b into the IV/V super domain separates these elements from the IDRR and the rest of domain V. It was demonstrated that replacement of the contemporary domain V with the Fermon domain V increased IRES activity in neuronal glioblastoma cells, while replacement of the Fermon domain V with a contemporary domain V leads to no statistically significant change in IRES activity [18]. This suggests that despite the drastic shifts in secondary and tertiary structures observed in the 2014 strain, domain V remains functionally active.

The polypyrimidine region between domain V and domain VI, identified as Y*_n_*-X*_m_*-AUG (where Y*_n_* is 8–10 pyrimidines, X*_m_* is 18–20 nucleotides and AUG is a cryptic start codon) [76,77], has been shown to contain an intrinsically disordered RNA region (IDRR) [41]. This sequence motif and its unstructured arrangement are supported by our models for both strains of EV-D68. This is the longest stretch of nucleotides that experiences no alterations in nucleotide sequence between the Fermon and 2014 strains. This high level of conservation could be expected given the IDRR’s role as the ribosomal landing pad for small subunit attachment and scanning [63,78]. Motifs that are important for proper recruitment of the ribosome, such as the UUUC motif at nucleotide 560 in the Fermon strain, are preserved in both the Fermon and 2014 tertiary structure models, which show the IDRR positioned on the outside of the molecule providing space for the ribosomal subunit to associate with the IRES [63].

Similar to domain III, the secondary structure of domain VI is highly conserved between the Fermon and 2014 strains, with only minor shifts in the base pairing of the internal loops. The cryptic AUG, which is utilized to promote viral translation and act as an alternative start site of translation of an upstream ORF (uORF) in many enterovirus species, but not in *Enterovirus D* [79,80], is located at position 587 in the Fermon strain and maintained within the 2014 strain.

Our work establishes comparative secondary structures and tertiary structure models for the 5′ UTR of the prototypic Fermon strain, as well as a 2014 outbreak strain of EV-D68 with a neurovirulent phenotype. Recent research that explores virulence determinants in EV-D68 shows that newly acquired neurovirulence is primarily, but not exclusively, a result of changes in the 5′ UTR [6,17,81]. These studies identify combinations of 5′ UTR and coding-region changes that correlate with neurovirulence. In a comprehensive genomic comparison of EV-D68 conducted by Zhang et al. [6], 21 genomic substitutions were identified as being diagnostic for neurovirulence. Eight of these substitutions were located in the 5′ UTR, with the others found in the structural proteins VP1, VP2 and VP3, as well as nonstructural proteins 2A, 2C and 3D. Interestingly, 12 of the 21 genomic substitutions identified in the study, including 5 in the 5′ UTR, changed into a nucleotide or amino acid identity also found in other neurovirulent enteroviruses. These viruses included enterovirus D70 (EV-D70), poliovirus (PV) and enterovirus A71 (EV-A71). Our 5′ UTR structural comparison also noted several examples where RNA elements in the EV-D68 2014 strain changed to a motif characteristic of PV.

We show that the structure of the 5′ UTR undergoes significant changes in overall organization in two strains of EV-D68 that differ in neurovirulence. These changes may underlie the shift in neurovirulence in contemporary EV-D68 strains. Specifically, the tertiary structure changes provide room to speculate about the potential for easy and efficient interaction of key functional 5′ UTR domains in the 2014 strain. The most active and important RNA elements in domains I, IV and V are clustered together with the pyrimidine-rich connecting region downstream of domain I and the IDRR downstream of domain V in the 5′ UTR of the 2014 strain. In contrast, the tertiary structure of the 5′ UTR from the Fermon strain is much less compact with greater separation of functional domains. Perhaps the potential for increased efficiency in the 2014 strain enables the virus to replicate efficiently in the cellular environment of neurons.

One pattern that emerges from our structural comparison involves changes that make the 2014 structure more similar to the poliovirus structure. While these changes are driven by relatively few sequence differences between the Fermon and 2014 5′ UTR, virulence determinants in enteroviruses are often localized to short regions or even single nucleotide positions [20,21,53,54,55,56,75,82]. Localizing virulence determinants in viral genomic RNA structures establishes promising targets for antiviral approaches. Such approaches have already produced a small molecule that inhibits EV-A71 replication by inducing a conformational change in domain II of the 5′ UTR [59]. Similar exploration of compounds that change the structure of EV-D68 virulence determinants could offer effective treatments for AFM caused by the virus.

## 5. Conclusions

EV-D68 is an enterovirus that was initially thought to only cause mild respiratory illness but during an outbreak in 2014 also became associated with AFM and has since been shown to be a causative agent of AFM [1,2,3,4,5,6,7,8,9,10]. One potential explanation for the appearance of neuropathogenicity is an increase in neurovirulence between historical and contemporary EV-D68 isolates. Determinants of virulence in the EV-D68 genome have been mapped to the 5′ UTR as well as the coding regions for the proteins VP1 and VP3 [6,17,18]. The 5′ UTR has been shown to be a determinant of virulence in other enteroviruses, and contemporary EV-D68 isolates display increased IRES activity of the 5′ UTR [18,19,20,21].

In this study, we utilized SHAPE-MaP to generate comparative secondary structures of the 5′ UTR between the prototypic Fermon strain and a 2014 outbreak isolate of EV-D68. These secondary structures then informed the formation of a predictive tertiary structure model of both strains using 3dRNAv2.0. The structures generated show shifts in the secondary structure of every domain within the 5′ UTR; however, within these domains, many elements shown to be important to the function of the 5′ UTR are maintained. The tertiary structure models show a shift from an extended conformation within the Fermon strain to a more compact globular structure within the 2014 strain. This change toward a more compact shape could increase the kinetic efficiency of the 5′ UTR by reducing the distance between sections that interact with one another. The structures we generated highlight domains that change drastically between the Fermon and 2014 strains and could serve as potential targets for therapeutics against EV-D68 disease. Small-molecule treatment targeting domain II of EV-A71 has been shown to be effective in attenuating virulence by stabilizing a tertiary structure that decreases the efficiency of viral multiplication [59]. With detailed knowledge of structural elements in the 5′ UTR of EV-D68, whether in strains that cause respiratory illness or in strains that cause AFM, similar therapeutics could be developed for the structural domains identified here in order to reduce the severity of and prevent disease in children infected with EV-D68.

## Figures and Tables

**Figure 1 viruses-15-00295-f001:**
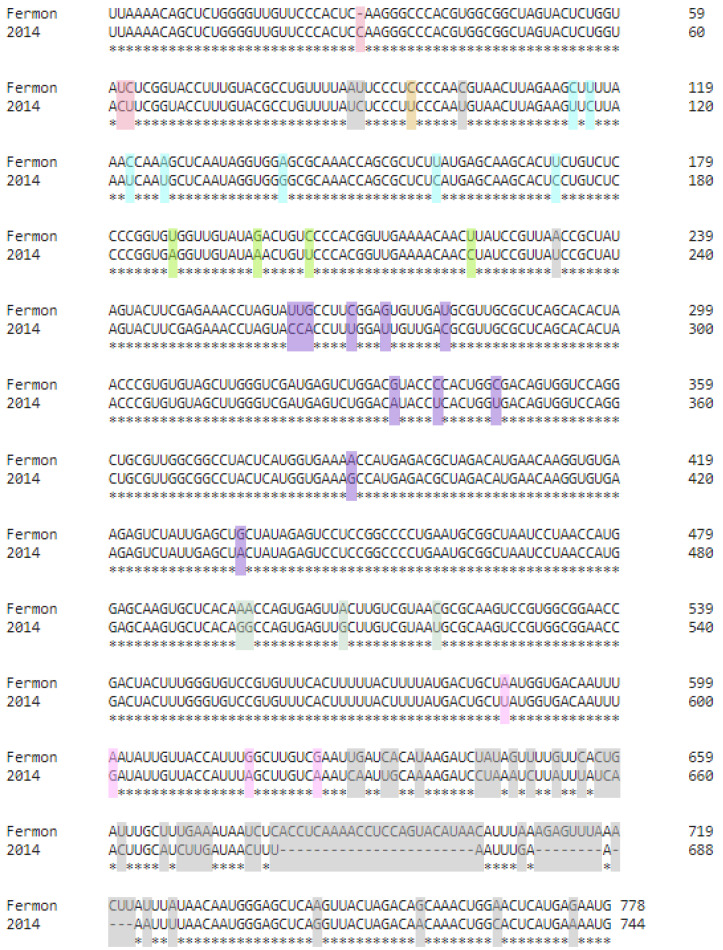
Multiple sequence alignment (MSA) of EV-D68 5′ UTRs. The sequence alignment shows base substitutions, insertions and deletions in the 2014 strain as compared to the original Fermon isolate. The MSA was generated using Clustal Omega. Using the Fermon strain as a reference, nucleotide alterations are colored based on the 5′ UTR domain: domain I = red, domain II = light blue, domain III = light green, domain IV = purple, domain V = sage green, domain VI = pink, and long-range interaction (LR) = khaki.

**Figure 2 viruses-15-00295-f002:**
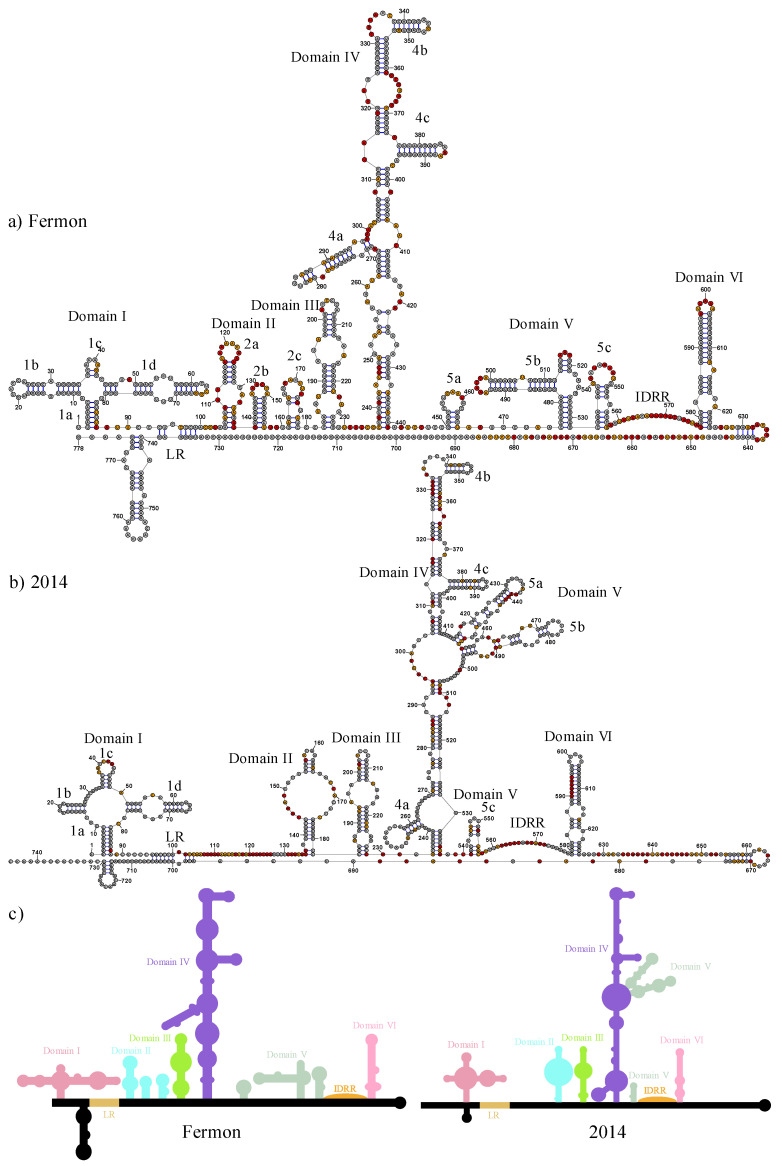
Secondary structure models of EV-D68 5′ UTRs. (**a**) and (**b**) represent the secondary structure of EV-D68 Fermon and 2014 strains, respectively. The shape reactivity of each nucleotide is indicated by colored shading. Grey shading indicates little reactivity, with reactivity values below 0.40. Orange shading indicates moderate reactivity, with reactivity values between 0.40 and 0.85. Red shading indicates high reactivity, with values greater than 0.85. Domains I through VI are labeled along with the long-range interaction (LR) and intrinsically disordered RNA region (IDRR). Line-art depictions with 5′ UTR of both the Fermon and 2014 strains are depicted in panel (**c**) and structural features are colored and labeled accordingly: domain I = red, domain II = light blue, domain III = light green, domain IV = purple, domain V = sage green, domain VI = pink, long-range interaction (LR) = khaki, and intrinsically disordered RNA region (IDRR) = orange.

**Figure 3 viruses-15-00295-f003:**
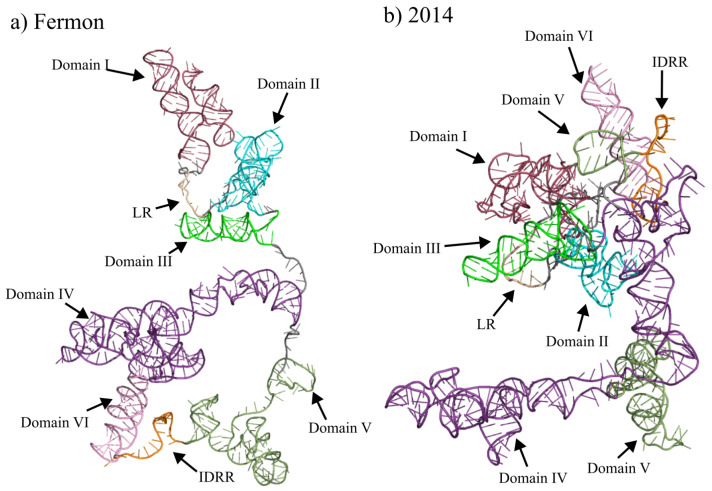
Tertiary structure models of the 5′ UTRs of EV-D68 Fermon and 2014 strains. Panels (**a**) and (**b**) represent the tertiary structure predictions generated using 3DRNAv2.0 for the 5′ UTRs of the Fermon and 2014 strains, respectively. Structural predictions were visualized and pseudocolored using PyMOL. Pseudocoloring follows the same coloring pattern as shown in panel (c) of Figure 2, and structures of interest are labeled as shown in panels of Figure 2.

**Figure 4 viruses-15-00295-f004:**
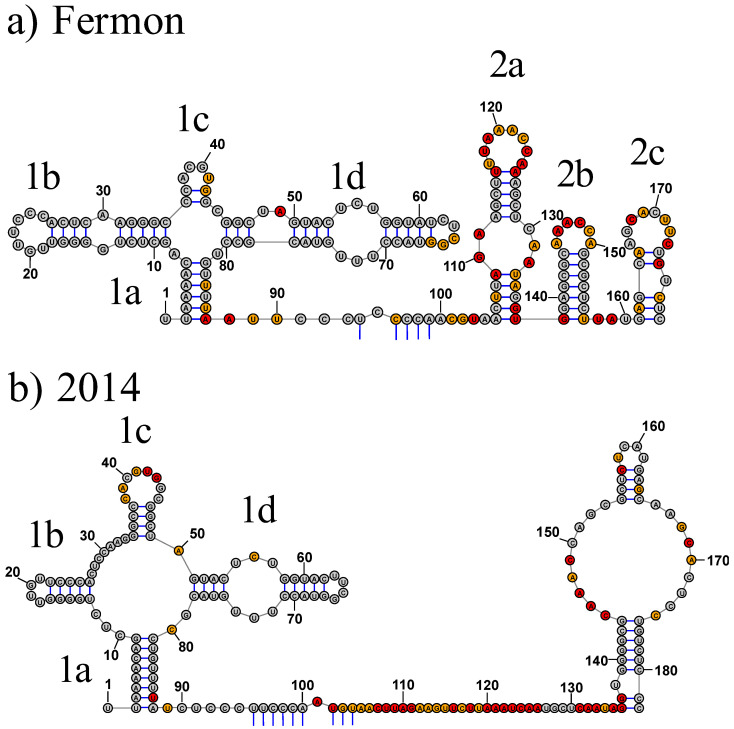
Secondary structure of domains I and II of EV-D68 5′ UTRs. (**a**) and (**b**) represent domains I and II of the 5′ UTR of the Fermon and 2014 strains of EV-D68, respectively. The shape reactivity of each nucleotide is indicated by colored shading. Grey shading indicates little reactivity, with reactivity values below 0.40. Orange shading indicates moderate reactivity, with reactivity values between 0.40 and 0.85. Red shading indicates high reactivity, with values greater than 0.85.

**Figure 5 viruses-15-00295-f005:**
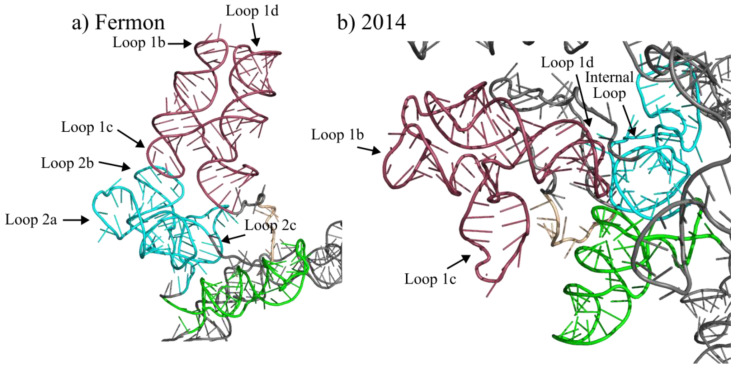
Tertiary structures of domains I, II and III of the Fermon and 2014 EV-D68 strains. Panels (**a**,**b**) represent the tertiary structure predictions generated using 3DRNAv2.0 for domains I, II and III of the 5′ UTRs of the Fermon and 2014 strains. Structural predictions were visualized and pseudocolored using PyMOL. The red regions represent domain I, the light blue regions represent domain II, and the light green region represent domain III. Domains and regions outside of I, II and III in this figure are colored grey.

**Figure 6 viruses-15-00295-f006:**
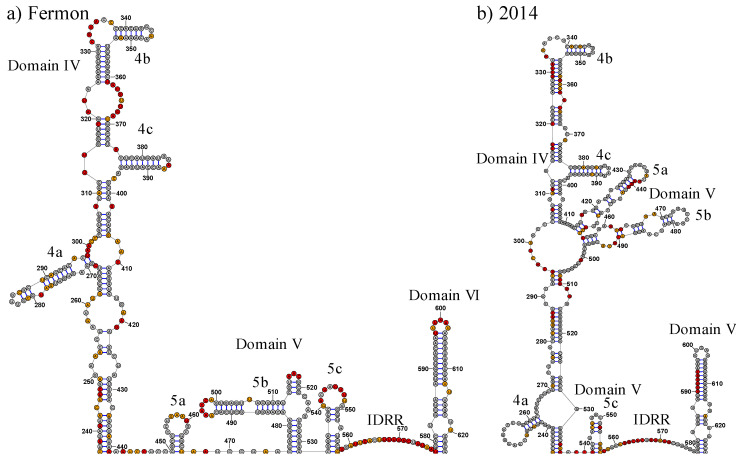
Secondary structures of domains IV and V of EV-D68 Fermon and 2014 Strains. (**a**) and (**b**) represent domains IV and V of the 5′ UTR of the Fermon and 2014 strains of EV-D68, respectively. The shape reactivity of each nucleotide is indicated by colored shading. Grey shading indicates little reactivity, with reactivity values below 0.40. Orange shading indicates moderate reactivity, with reactivity values between 0.40 and 0.85. Red shading indicates high reactivity, with values greater than 0.85.

**Figure 7 viruses-15-00295-f007:**
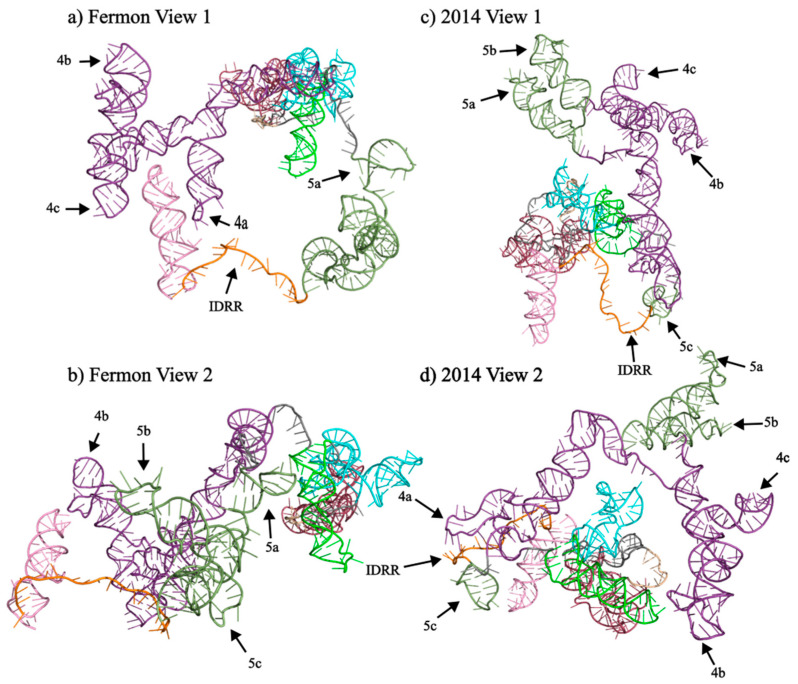
Tertiary structures domains IV and V of EV-D68 Fermon and 2014 Strains. Panels (**a**–**d**) represent the tertiary structure predictions generated using 3DRNAv2.0 for domains IV and V of the 5′ UTRs of the Fermon and 2014 strains. Structural predictions were visualized and pseudocolored using PyMOL. The coloring of the domains follows the pattern set in Figure 1. Structures of interest are labeled within each panel. Panels (**b**) and (**d**) represent the Fermon and 2014 5′ UTR structures, respectively, viewed from a different angle.

**Table 1 viruses-15-00295-t001:** Homologous Sequences Used in RNA Folding of the Fermon and 2014 strains.

	Viral Isolate	Accession Number
Fermon Homologous Sequences	USA/MN/1989-23220	MN240496.1
	USA/MO/2000-23221	MN240497.1
	USA/CA/1962-23234	MN240508.1
	TTa-08-Ph561	LC629436.1
	USA/WI/2006-23109	MN240491.1
	USA/AL/2007-23110	MN240492.1
	USA/KS/2007-23111	MN240493.1
	USA/AK/2008-23112	MN240494.1
	USA/MD/2009-23114	MN240495.1
	USA/TX/2002-23222	MN240498.1
2014 Homologous Sequences	CF425314-FRA-2018	MT789741.1
	CF194006-FRA-2016	MT791930.1
	CF183054-FRA-2016	MT791933.1
	2016-R3936	MH341711.1
	2016-R4010	MH341712.1
	2016-R4609	MH341715.1
	TW-00932-2014	KT711081.1
	TW-00821-2014	KT711084.1
	TW-00928-2014	KT711085.1
	TW-02512-2014	KT711086.1

## Data Availability

FASTQ files from SHAPE-MaP analysis are available by request by contacting wtapprich@unomaha.edu.

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
