# Peer review of "RNA Structure in the 5′ Untranslated Region of Enterovirus D68 Strains with Differing Neurovirulence Phenotypes"

_viruses, 2023, doi:10.3390/v15020295_

Round 1

Reviewer 1 Report

I have no particular criticism to make about this paper, which reports very interesting and sound data. Of course, one is left to wonder how the changes in sequence and structure of the virus 5'UTRe exactly influence its ability to replicate in a given cell lineage. Perhaps the authors may be willing to speculate a bit more about this point.

Reviewer 2 Report

In this manuscript, the authors analyzed the 5'UTR structure of the pre-2014 EV-D68 and the 2014 pandemic EV-D68. From the results, it was found that the 5'UTR of the 2014 pandemic EV-D68 had several nucleotide mutations, but this mutation was found in the 2-node structure and the triple structure analysis revealed that the domainI-VI of the 5'UTR was reassembled into a completely different structure than the original 5'UTR of EV-D68. These structural changes may be helpful for the EV-D68 translation, and the reorganization of these domainI-VI structures leads to a closer distance between them, which may also improve the kinetic efficiency of the 5'UTR. Such a study will help us to develop a small molecule therapeutic approach for 5'UTR of EV-D68. However, some issues need to be clarified, such as:

1. 1. We know that the authors want to analyze whether the 5'UTR structure of EV-D68 is also changed, because there is an IRES structure on the 5'UTR of EV-D68 that can help the virus to translate. However, since EV-D68 itself also has a coding region, and the 3D of the virus will help the virus replication, 2B and 2BC will change the permeability of the cell membrane to regulate the virus replication. Therefore, I would like to ask whether the RNA in these coding regions of EV-D68, which broke out in the pandemic in 2014, also had mutations that changed the structure of the viral protein, resulting in increased infection of EV-D68. Will the authors continue to study the coding regions of EV-D68 in the 2014 pandemic?

2. In the conclusion, there is a paragraph stating that analyzing the difference in the 5'UTR structure between pre-2014 EV-D68 and 2014 pandemic EV-D68 could be useful for the treatment of EV-D68, and that small molecule therapies could be developed for these 5'UTR domains, but I would like to ask whether these small molecule therapies are mainly for EV-D68 before 2014, or for EV-D68 that exploded in pandemic in 2014, or to develop small molecule therapies with the same inhibitory effect for EV-D68 of these two different genetic sequences. Can the authors provide more details regarding this issue?

Round 2

Reviewer 2 Report

In the response to my first question, the authors mentioned in lines 565-569 of the discussion that the 5'UTR of EV-A71 from the 2014 outbreak had 21 nucleotide mutations, and that the nucleotide positions of these mutations were similar to those of the 5'UTRs of other neurotoxic enteroviruses, However, I would like to ask the authors what are the neurotoxic enteroviruses that have similar positions to the 5'UTR of EV-A71 in the 2014 outbreak?  Would the authors please list them and provide literature references?

The authors suggest that the 5'UTR structure of EV-A71 in the 2014 outbreak is different due to nucleotide changes, but I would like to ask the authors how they can be sure that the 5'UTR structure change of EV-A71 is directly related to the elevated neurovirulence of the virus itself, whether there is any literature available, or the authors will do relevant experiments to verify this later?
